# Specific Detection of African Swine Fever Virus Variants: Novel Quadplex Real-Time PCR Assay with Internal Control

**DOI:** 10.3390/microorganisms13030615

**Published:** 2025-03-07

**Authors:** Lihua Wang, Yuzhen Li, Xirui Zhang, Rachel Madera, Homer Pantua, Aidan Craig, Nina Muro, Danqin Li, Jamie Retallick, Franco Matias Ferreyra, Quang Lam Truong, Lan Thi Nguyen, Jishu Shi

**Affiliations:** 1Center on Biologics Development and Evaluation, College of Veterinary Medicine, Kansas State University, Manhattan, KS 66506, USA; yuzhen@vet.k-state.edu (Y.L.); xiruizhang@ksu.edu (X.Z.); rachelmadera@vet.k-state.edu (R.M.); aidan3@vet.k-state.edu (A.C.); nina14@vet.k-state.edu (N.M.); danqinli@vet.k-state.edu (D.L.); 2Department of Anatomy and Physiology, College of Veterinary Medicine, Kansas State University, Manhattan, KS 66506, USA; 3BioAssets Corporation, Santo Tomas 4234, Batangas, Philippines; homer.pantua@bioassets.com.ph; 4Department of Diagnostic Medicine and Pathobiology, College of Veterinary Medicine, Kansas State University, Manhattan, KS 66506, USA; retallick@vet.k-state.edu (J.R.); francomf@vet.k-state.edu (F.M.F.); 5Key Laboratory of Veterinary Biotechnology, Faculty of Veterinary Medicine, Vietnam National University of Agriculture, Gia Lam, Ha Noi 12406, Vietnam; tqlam@vnua.edu.vn (Q.L.T.); nguyenlan@vnua.edu.vn (L.T.N.)

**Keywords:** African swine fever, variants, real-time PCR, internal control, sensitive, specific

## Abstract

African swine fever (ASF), a highly contagious and lethal viral disease, continues to devastate the global swine industry. The emergence of ASF virus (ASFV) variants with varying genomic deletions poses significant challenges for ASF control. This study presents a novel, sensitive, and reliable quadplex real-time PCR assay for detecting ASFV variants lacking key genes (*I177L*, *EP402R*, and *MGF360-14L*), either individually or in combination. The assay targets conserved regions within these genes, ensuring broad coverage of diverse ASFV genotypes. A porcine *beta-actin* (*ACTB*) internal control was incorporated to minimize false-negative results. Optimization and evaluation using spike-in tests demonstrated high sensitivity, with a limit of detection (LOD) ranging from 1 to 10 plasmid copies or 0.1 TCID_50_ of ASFV isolates per reaction. No cross-reactivity was observed when testing serum samples from pigs infected with other common swine viruses. Further validation across a diverse panel of samples, including those from naturally ASFV-infected field pigs (*n* = 54), experimentally ASFV-infected pigs (*n* = 50), PBS-inoculated pigs (*n* = 50), ASFV-free field pigs (*n* = 100), and feral pigs (*n* = 6), confirmed 100% specificity. This robust assay provides a valuable tool for rapid and accurate ASF surveillance and control efforts, facilitating the timely detection and mitigation of outbreaks caused by emerging ASFV variants.

## 1. Introduction

African swine fever (ASF) is a highly contagious and lethal viral disease affecting both domestic and wild pigs [1]. First identified in East Africa in the early 1900s, it has plagued the global swine industry for over a century [2]. Classified as a notifiable disease by the World Organization for Animal Health (WOAH), ASF has caused substantial economic losses due to pork shortages, trade restrictions, and costly eradication efforts [3,4]. Currently, ASF remains widespread in sub-Saharan Africa, parts of West Africa, and Sardinia and continues to spread in European, Asian, Pacific, and Caribbean regions [4,5]. The lack of widely available vaccines and effective treatments poses a significant challenge to ASF control and eradication efforts.

African swine fever virus (ASFV), the causative agent of ASF, belongs to the genus *Asfarvirus*, within the *Asfarviridae* family [6]. It infects all members of the *Suidae* family, including domestic pigs, wild boars, warthogs, and bush pigs [7,8]. Clinical signs in infected pigs vary widely, ranging from highly fatal peracute infections to milder chronic forms, depending on the virulence of the virus strain and the age of the pig [9]. Highly virulent strains can cause nearly 100% mortality, while moderately virulent strains typically result in mortality rates between 30% and 70% [10]. The ASFV genome is large and complex, comprising double-stranded DNA containing over 150 open reading frames (ORFs) [6,10]. Over 100 viral proteins have been identified, many with unknown functions [6,7,8,9,10,11]. ASFV has been classified into 24 genotypes (I–XXIV) based on the sequence of its major capsid protein, p72 [12]. While all 24 genotypes circulate in Africa, only genotypes I and II have been detected outside the continent [8,13]. Notably, the highly virulent strains currently affecting parts of Europe and Asia, as well as the Caribbean, belong to genotype II [4,8,14,15,16,17]. Furthermore, highly lethal recombinant ASFVs of genotypes I and II have been detected in Asia [18,19].

The lack of effective control tools severely hampers ASF prevention and control. Traditional vaccination approaches, including those using inactivated viruses, infected cell extracts, and subunit vaccines targeting specific viral proteins, have failed to consistently induce protective immunity [20,21,22,23]. However, live-attenuated vaccines (LAVs) generated by the targeted removal of virulence genes show promise. The multigene family (MGF) and *I177L* are two successful targets for generating ASF LAVs. Two LAVs (AVAC ASF Live and NAVET-ASFVAC) have been approved for commercial use in Vietnam. AVAC ASF Live is based on the ASFV-G-ΔMGF strain with six MGF gene deletions. NAVET-ASFVAC is based on the ASFV-G-ΔI177L strain, developed by partially deleting the *I177L* gene [24,25,26,27]. The *EP402R* gene, encoding the CD2v protein essential for hemadsorption, is often co-deleted with other virulence genes to enhance vaccine safety or serve as a DIVA marker [8,28,29,30,31,32]. The low-virulence variants found in Asia in 2021 harbored deletions in MGF genes and *EP402R* [33]. The emerging of naturally occurring, non-hemadsorbing (Non-HAD) ASFVs is related to *EP402R* gene deletion [34,35,36]. These variants are milder but highly transmissible, which creates new challenges for ASF prevention and control. An effective and timely response requires not only the detection of wild-type ASFVs but also the precise identification of circulating ASFV variants, including gene-deleted LAVs and naturally occurring ASFV variants.

There are several approved diagnostic tests for ASFV. The WOAH recommends virus isolation, fluorescent antibody (FAT) testing, and both real-time PCR and conventional PCR [37]. Although virus isolation is the gold standard, its time-consuming nature limits its applicability for real-time disease monitoring. FAT exhibits reduced sensitivity in subacute and chronic ASF. PCR-based techniques, known for their sensitivity, specificity, rapidity, and versatility, are crucial for ASFV detection and are currently the most widely used approaches. They enable early detection of the ASFV genome in tissue, bloodand serum samples [38]. Numerous commercial PCR kits are available, including the INgene qPPA (Gold Standard Diagnostics, Davis, CA, USA), ASFV virus type (Indical Bioscience, Leipzig, Germany), ID Gene ASF Duplex (IDvet, Grabels, France), RealPCR ASFV (IDEXX, Westbrook, ME, USA), SwineFever combi (Gerbion, Kornwestheim, Germany), ViroReal kit ASFV virus (Ingenetix, Wien, Austrilia), and Kylt ASF Real-Time PCR kits (Anicon, Höltinghausen, Germany) [39]. The WOAH-certified VetMAX™ African Swine Fever Virus Detection kit (Thermo Fisher Scientific, Waltham, MA, USA), which underwent the WOAH Biological Standards Commission’s registration procedure, is recommended for detecting ASF virus in blood, serum, and tissues from both domestic and wild swine [40]. However, while these kits are generally reliable for detecting wild-type ASFV, they are not suitable for detecting gene-deleted ASFV variants.

Multiplex PCR, similar in principle to conventional PCR, allows for the simultaneous detection of multiple viruses without cross-reactivity [41]. We previously developed a multiplex real-time PCR panel for detecting 12 common swine viruses [42]. In this study, we developed a novel quadplex real-time PCR assay with an internal control. This assay aims to directly detect ASFV variants lacking the *I177L*, *EP402R*, and *MGF360-14L* genes, either individually or in combination, which may arise naturally or through vaccination. It will be a valuable tool for detecting and differentiating gene-deleted ASFV variants from wild-type strains.

## 2. Materials and Methods

### 2.1. Viruses and Porcine Serum Samples

The ASFV VNUA-ASFV-05L1 strain (genotype II), ASFV Georgia strain (genotype II), ASFV OURT88/1 strain (genotype I), classical swine fever virus (CSFV) Alfort strain, and CSFV vaccine C-strain are maintained in a BSL-3 laboratory at Kansas State University (KSU). Porcine reproductive and respiratory syndrome virus (PRRSV) strains (VR-2332, NADC-20, JXA1-R, and 1-4-4L1C) and porcine circovirus type 2 (PCV2b) are kept in Dr. Jishu Shi’s laboratory at KSU. The PRV Bartha-K61 strain was kindly provided by Dr. Lynn W. Enquist (Princeton University) and is kept in Dr. Jishu Shi’s laboratory at KSU. These viruses were used for preparing the standards and performing sensitivity and specificity tests for the evaluation of the quadplex real-time PCR assay.

This study utilized the following serum panels for evaluating the sensitivity and specificity of the quadplex real-time PCR assay:(i)ASFV-negative pig sera: serum samples from pigs inoculated with phosphate-buffered saline (PBS, pH7.4, Thermo Scientific, Bridgewater, NJ, USA) (*n* = 50).(ii)Experimentally ASFV-infected pig sera: serum samples from pigs infected with the ASFV VNUA-ASFV-05L1 strain and were confirmed as ASFV positive when tested by standard ASFV real-time PCR [42] (*n* = 50).(iii)Naturally ASFV-infected pig sera: serum samples from naturally ASFV-infected field domestic pigs in an ASFV-epidemic country (the Philippines) (*n* = 54).(iv)ASFV-free field domestic pig sera: serum samples from pigs on local farms in Kansas, USA (*n* = 100).(v)Feral pig sera: serum samples from feral pigs caught in Kansas (collaboration with USDA APHIS Wildlife Services, Kansas Wildlife Services, USA) (*n* = 6).(vi)Other common swine virus-infected pig sera: serum samples from pigs infected with CSFV (*n* = 50), PRRSV (*n* = 50), PRV (*n* = 10), and bovine viral diarrhea virus (BVDV, *n* = 4).

Panels (ii) and (iii), as well as serum samples from pigs infected with CSFV, are maintained in a BSL-3 laboratory at KSU. All other serum samples are kept in Dr. Jishu Shi’s laboratory at KSU.

### 2.2. Construction Databases, Sequence Analysis, and Design of Primers and Probes

All currently available sequences of the ASFV *I177L*, *EP402R*, and *MGF360-14L* genes and genomes containing these genes from GenBank, online ASFV sequence databases (https://asf-referencelab.info/sequence-database/ and http://asfvdb.popgenetics.net/, accessed on 1 June 2024), and other published sources were downloaded. Sequence databases for the *I177L* (*n* = 235), *EP402R* (*n* = 263), and *MGF360-14L* (*n* = 236) genes were constructed by combing individual *I177L*, *EP402R*, and *MGF360-14L* gene sequences with sequences derived from ASFV whole genomes. These databases included genotypes I, II, III, IV, V, VII, VIII, IX, X, XX, and XXII. The alignment of nucleotide sequences was performed by using the ClustalX program, version 2.1 [43]. Phylogenetic and evolutionary analyses were conducted using MEGA 11 [44]. Neighbor-joining (NJ) phylogenetic trees were generated using the p-distance model. The robustness of the branching was evaluated by bootstrapping using 1000 replications.

The primers and probes were designed in the most conserved region of the target gene that was identified from multiple sequence alignments to cover 100%, as calculated by percentage, of the total sequences that matched at least one forward primer, one reverse primer, and one probe sequence for each sequence. The predicted amplicon size was limited to 100–150 bp for each primer pair to potentially increase the reaction sensitivity. Primers and probes were checked for potential secondary structures and dimer formations prior to synthesis with the NCBI tool Primer-Blast (https://www.ncbi.nlm.nih.gov/tools/primer-blast/index.cgi (accessed on 1 July 2024)). All oligonucleotides were synthesized by Integrated DNA Technologies, Inc. (Coralville, IA, USA). Information on the primer and probe sequences, amplicon sizes, targeted genes, and numbers of available sequences used for the design for each virus is outlined in Table 1. To detect the porcine housekeeping gene *beta-actin* (*ACTB*), we incorporated previously published primers (forward: 5′-GACCTGACCGACTACCTCATG-3′; reverse: 5′-TCTCCTTGATGTCCCGCAC-3′) and an established probe (CY5-CTACAGCTTCACCACCACGGC-BHQ2), which demonstrated excellent diagnostic performance, with 100% coverage over 69 available sequences, and did not interfere with the multiplex reaction [45].

### 2.3. Preparation of Standard Plasmids and Optimization of Amplification Conditions

The target genes *I177L*, *EP402R*, and *MGF360-14L* were synthesized and cloned into pUC57 vectors by GenScript (Piscataway, NJ, USA). The plasmids were transformed into NEB^®^ 10-beta Competent *E. coli* (New England Biolabs, Ipswich, MA, USA), purified using a Qiagen plasmid Midi Kit (Qiagen, Germantown, MD, USA), and quantified with a NanoPhotometer P330 instrument (Implen, Schatzbogen, München, Germany). The plasmid copy number was calculated using the following formula: Plasmid copies/µL = (6.02 × 10^23^) × (ng/µL × 10^−9^)/(bp of plasmid × 660). Each plasmid was adjusted to a concentration of 10^10^ copies/µL and serially diluted tenfold with serum samples from a healthy donor pig to generate individual standard curves and determine the limit of detection (LOD).

DNA extraction was performed using an automated KingFisher™ Flex Purification System (ThermoFisher Scientific, Waltham, MA, USA) with a MagMAX™ Viral/Pathogen Nucleic Acid Isolation Kit (ThermoFisher Scientific, MA, USA) for 200 µL samples following the manufacturer’s instructions. Real-time PCR was performed using a CFX96™Touch™ Real-Time PCR Detection System (Bio-Rad, Hercules, CA, USA). Several multiplex reaction buffers, including an iQ™ Multiplex Powermix kit (Bio-Rad, CA, USA), a QIAGEN Multiplex PCR Kit (Qiagen, MD, USA), Platinum™ Multiplex PCR Master Mix (Thermo Fisher Scientific, MA, USA), Multiplex PCR 5X Master Mix (New England Biolabs, MA, USA), and a Path-ID Multiplex One-Step Real-Time PCR Kit (Thermo Fisher Scientific, MA, USA), were evaluated according to their respective manufacturers’ instructions. Primer and probe final concentrations were set at 400 nM and 200 nM, respectively. Thermocycling parameters were adjusted according to each manufacturer’s instructions (Table 2). All the experiments were performed in triplicate.

### 2.4. Analytical Sensitivity and Specificity Evaluation by Spiking Experiments

To assess analytical sensitivity, 10^5^ TCID_50_ (50% tissue culture infectious dose) of cell-cultured ASFV strains VNUA-ASFV-05L1 (genotype II), Georgia (genotype II), and OURT88/1 (genotype I) were spiked into serum samples from a healthy donor pig. These spiked samples were serially diluted tenfold. To evaluate analytical specificity, 10^5^ TCID_50_ of the CSFV Alfort strain, CSFV C-strain, PRRSV strains (VR-2332, NADC-20, JXA1-R, 1-4-4L1C), PCV2b strain, and PRV Bartha-K61 strain were spiked into serum samples from a healthy donor pig. Nucleic acid extraction was performed as described above. Quadplex real-time PCR was conducted using the Path-ID Multiplex One-Step Real-Time PCR Kit, incorporating a reverse transcription step at 48 °C for 10 min, followed by inactivation and denaturation at 95 °C for 5 min. Subsequent PCR cycles involved 45 cycles of denaturation at 95 °C for 15 s and annealing/extension at 60 °C for 45 s (Table 2). All the experiments were performed in triplicate and compared with the standard singular ASFV real-time PCR [42]. Data analysis was performed using Bio-Rad CFX Maestro software 1.1 (Bio-Rad, CA, USA).

### 2.5. Validation of Quadplex Real-Time PCR with Experimental and Field Samples

To validate the quadplex real-time PCR assay with experimental clinical samples, serum samples from panels (i), (ii), and (vi) were tested to evaluate its diagnostic sensitivity and specificity. To validate the quadplex real-time PCR assay with field clinical samples, serum samples from panels (iii), (iv), and (v) were tested. Nucleic acid extraction was performed as described above. Quadplex real-time PCR was conducted using the Path-ID Multiplex One-Step Real-Time PCR Kit, incorporating a reverse transcription step at 48 °C for 10 min, followed by enzyme inactivation and denaturation at 95 °C for 5 min. Subsequent PCR cycles consisted of 45 cycles of denaturation at 95 °C for 15 s and annealing/extension at 60 °C for 45 s. All experiments were performed in triplicate, and the results were compared with those obtained using the standard singular ASFV real-time PCR [42].

### 2.6. Statistical Analysis

All data were obtained from three independent experiments. The significance of the correlation coefficient between the quadplex real-time PCR and the standard singular real-time PCR was determined using a *t*-test in SPSS Statistics for Windows, version 25.0 (IBM Corp., NY, USA). Box-and-whisker plots were created in Excel 2019. Sensitivity and specificity analyses were performed using the web-based MedCalc statistical software 23.1.7 (https://www.medcalc.org/calc/diagnostic_test.php, accessed on 21 January 2025).

## 3. Results

### 3.1. Database for Sequence Alignment and Design of Primers and Probes

To construct the *I177L*, *EP402R*, and *MGF360-14L* gene databases, we obtained 235, 263, and 236 sequences, respectively. These sequences represented the *p72* gene-based genotypes I, II, III, IV, V, VII, VIII, IX, X, XX, and XXII (Table 1). While the majority of sequences were from genotypes I and II, a smaller number originated from other genotypes. Genotypes VI, XI–XIX, XXI, XXIII, and XXIV were excluded due to a lack of available sequences for these genotypes in all three gene databases. Phylogenetic analysis of the *I177L*, *EP402R*, and *MGF360-14L* gene sequences revealed that the clades corresponding to different genotypes were consistent with those identified using the *p72* gene. The genetic diversity of these three genes varied, with *MGF360-14L* exhibiting the highest level of diversity, followed by genes *EP402R* and *I177L*, as evidenced by the horizontal branch lengths in the phylogenetic tree (Figure 1).

Our novel quadplex real-time PCR assay is designed to detect *I177L*, *EP402R*, and *MGF360-14L* gene-deleted ASFV variants using a combination of probes and a porcine *ACTB* internal control (Table 3). Wild-type ASFVs will yield positive results for all probes, while gene-deleted ASFV variants will show negative results for one to three probes. This strategy enables differentiation between infections caused by wild-type and gene-deleted ASFV variants. To ensure comprehensive coverage, primers were selected from the most conserved regions of the target genes. If a single primer/probe could not cover all sequences in a given database, multiple primers/probes were chosen to achieve 100% coverage (Table 1). Primers and probes were designed within a narrow annealing temperature range and a length of 100–150 bp to optimize the compatibility and sensitivity of the multiplex reactions.

### 3.2. Optimization of Quadplex Real-Time PCR Through Standard Plasmid Spiking Experiments Using Various Multiplex Reaction Buffers

To evaluate the performance of multiplex reaction buffers for the quadplex real-time PCR assay, we serially diluted a standard plasmid containing the target gene with serum samples from a healthy donor pig. These diluted samples were then tested with five different multiplex reaction buffers. The Path-ID Multiplex Real-Time PCR reaction buffer exhibited the optimal performance, achieving an LOD of 1–10 plasmid copies, with corresponding cycle threshold (Ct) values ranging from 35 to 38. The standard curves demonstrated high correlation coefficients (R^2^ ≥ 0.98) and efficient PCR amplification (E = 104–105%). While the other four tested buffers also yielded acceptable results (R^2^ ≥ 0.94, E ≥ 80%), their LODs were slightly lower (10–100 plasmid copies) (Table 4). Based on these findings, Path-ID Multiplex One-Step Real-Time PCR reaction buffer was selected for our quadplex real-time PCR system. Consistent with previous findings [45], the amplification of the internal control *ACTB* gene did not exhibit interference with target gene amplification. Its Ct values remained stable around 28 in all tested samples, and it did not influence the standard curves.

### 3.3. Analytical Sensitivity and Specificity of Quadplex RT-PCR in Virus-Spiked Serum Samples

Quadplex real-time PCR demonstrated excellent sensitivity when applied to ASFV-spiked pig serum samples. The LOD was determined to be 0.1 TCID_50_ for both genotype I and genotype II viruses across all three target genes, aligning with the performance of standard singular real-time PCR assay (Table 5). The strong correlation between quadplex real-time PCR and standard singular real-time PCR, with correlation coefficients ranging from 0.97 to 0.99, further supports the sensitivity of the quadplex real-time PCR assay. The internal control gene *ACTB* was consistently amplified, with Ct values around 28, indicating no interference with target gene amplification in all tested samples.

To evaluate specificity, the quadplex real-time PCR was applied to pig serum samples spiked with high quantities (10^5^ TCID_50_) of other common swine viruses, including CSFV, PRRSV, PCV, and PRV (Table 5). No cross-reactivity was observed, confirming the specificity of the assay.

### 3.4. Diagnostic Sensitivity and Specificity of Quadplex Real-Time PCR in Serum Samples from Experimentally Infected Pigs

To further evaluate the quadplex real-time PCR assay, we used serum samples collected from ten pigs infected with ASFV VNUA-ASFV-05L1 (genotype II) from 7 DPI (days post-infection) to the day they were euthanized (between 8 and 16 DPI). This virus was isolated from a domestic pig during an ASF outbreak in Northern Vietnam in 2020 and caused typical clinical signs of acute ASF [46]. All 50 samples that tested positive by standard singular ASFV real-time PCR were positive for the targets included in quadplex real-time PCR. Quadplex RT-PCR showed similar Ct values, ranging from 19 to 40, to those of the standard singular ASFV real-time PCR (Figure 2, Table 6).

We further tested our quadplex real-time PCR with various serum sample categories from pigs experimentally infected with other common swine viruses. The results are encouraging (Table 6). The quadplex real-time PCR did not show any false positives, meaning it had 100% specificity for detecting ASFV in these samples.

### 3.5. Performance of Quadplex Real-Time PCR in Field Serum Samples

To evaluate the performance of the quadplex real-time PCR with field samples, we analyzed sera from three sources: (1) naturally ASFV-infected domestic pigs in the Philippines (an ASFV-endemic country), (2) domestic pigs from local farms in Kansas, and (3) feral pigs captured in Kansas. The quadplex real-time PCR demonstrated ideal results, yielding negative results for all ASFV-free field samples from domestic pigs in Kansas (*n* = 100). All positive samples from ASFV-infected pigs in the Philippines, previously identified as positive by the standard singular ASFV real-time PCR [42], also tested positive with our quadplex real-time PCR, confirming 100% specificity. Moreover, serum samples from feral pigs (*n* = 6) were all negative using the quadplex real-time PCR, further supporting 100% specificity when compared to the standard singular ASFV real-time PCR (Table 7).

## 4. Discussion

ASF remains a significant global threat due to the complex nature of the virus, the lack of a widely available effective vaccine, and persistent knowledge gaps despite recent research progress [4]. Currently, ASF control still primarily relies on early detection and robust biosecurity measures. However, the emergence of gene-deleted, low-virulence ASFV variants, often exhibiting milder disease and reduced mortality, presents new challenges for diagnosis and control [33,34,35,36]. Therefore, developing rapid, sensitive, and accurate diagnostic methods for detecting circulating ASFVs, including wild-type ASFVs, gene-deleted LAVs, and naturally occurring ASFV variants, is crucial for implementing timely and targeted interventions to prevent further spread and mitigate their impact on pig populations.

Real-time PCR is a rapid and sensitive method for detecting ASFV DNA in clinical samples. Numerous highly sensitive and specific real-time PCR assays have been developed and validated for ASFV diagnosis worldwide [47,48]. In addition, multiplex real-time PCR has been employed to improve efficiency by detecting multiple targets simultaneously. For example, multiplex real-time PCR is used to distinguish ASFV genotypes I and II [49] and to simultaneously detect multiple swine viruses [42,50]. These PCR assays, while efficient, generally cannot differentiate between ASFV variants and wild-type strains. This study presents a novel, sensitive, and specific quadplex real-time PCR assay for the direct detection of ASFV variants lacking the *I177L*, *EP402R*, and *MGF360-14L* genes, either individually or in combination. This assay significantly improves detection efficiency by simultaneously screening for multiple variants and has the ability to differentiate between *I177L*, *EP402R*, and *MGF360-14L* gene-deleted ASFV variants and wild-type ASFVs (Table 3 and Table 4). It can significantly improve turnaround time for suspected samples and allows for faster implementation of control measures, even before full characterization at the reference lab. It may also be particularly useful in resource-limited settings where rapid initial screening is critical.

The assay exhibited remarkable specificity (100%), ensuring exclusive amplification and detection of the targeted ASFV viral nucleic acids. High sensitivity, with a limit of detection (LOD) ranging from 1 to 10 plasmid copies or 0.1 TCID50 of ASFV isolates (for both genotypes I and II) per reaction for all three genes, enabled the detection of even low viral loads, crucial for early stage infection diagnosis. Standard curves demonstrated high correlation coefficients (R^2^ ≥ 0.98) and efficient PCR amplification (E = 104–105%). The porcine *ACTB* gene, a housekeeping gene present in most sample types, served as an internal control, eliminating the need for additional preparation or inoculation steps [45]. Consistent amplification of the porcine *ACTB* internal control with Ct values of around 28 across all pig samples indicated effective monitoring of extraction efficiencies, no interference with target gene amplification, and robust diagnostic performance.

The developed quadplex real-time PCR assay was employed to analyze both experimental and field clinical samples. Compared to the standard singular ASFV real-time PCR, it exhibited excellent sensitivity. All samples that tested positive with the standard singular assay also yielded positive results for the targets included in the quadplex assay. Moreover, quadplex real-time PCR demonstrated comparable Ct values (ranging from 19 to 40) to the standard assay (Figure 2, Table 6). Importantly, the quadplex assay showed 100% specificity across naturally ASFV-infected and ASFV-free field samples, including those from feral pigs, indicating robust field diagnostic performance. These results underscore the potential of this assay as a valuable field-deployable tool for identifying and monitoring ASFV variants in wild pig populations. A potential limitation of this assay lies in the possibility of mismatches due to evolving ASFV strains, including hypothetical variants with significant alterations or deletions within the targeted genes. To mitigate this risk, we will continuously monitor published and GenBank-deposited ASFV sequences, ensuring that our primers and probes remain up-to-date and reliable. Furthermore, comprehensive field validation studies, encompassing diverse sample types and environmental conditions, are planned to solidify these promising findings. We will report the findings soon.

## 5. Conclusions

This study successfully developed a sensitive and accurate quadplex real-time PCR assay with an internal control for the detection and differentiation of ASFV variants harboring deletions in the *I177L*, *EP402R*, and *MGF360-14L* genes from wild-type ASFV strains. This robust assay, characterized by high sensitivity and specificity, demonstrates significant potential for the effective detection and investigation of ASFV variants in field clinical samples.

## Figures and Tables

**Figure 1 microorganisms-13-00615-f001:**
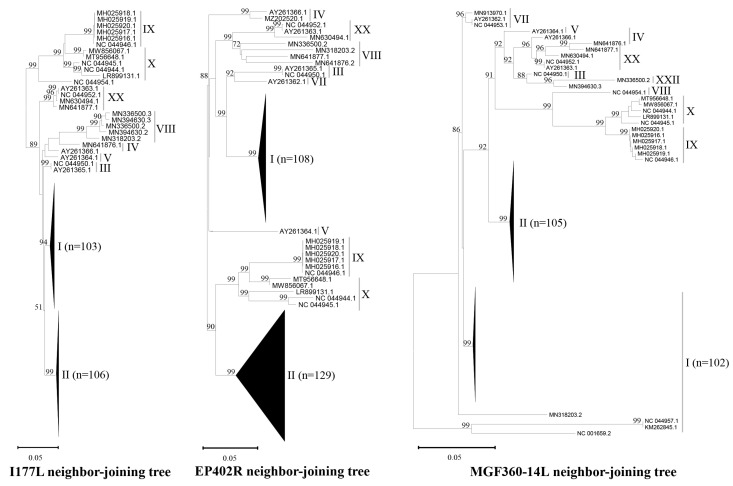
Phylogenetic analysis of sequences of *I177L*, *EP402R*, and *MGF360-14L* from databases. The neighbor-joining trees were constructed using MEGA11 [44]. Bootstrap support values (1000 replicates) are shown above branches. The genetic distances were calculated using the p-distance method. Horizontal branch lengths are proportional to genetic distance, and vertical branch lengths have no significance. Sequences are identified by using GenBank numbers. The numbers I to V, VII to X, XX, and XXII ndicate genotypes I to V, VII to X, XX, and XXII.

**Figure 2 microorganisms-13-00615-f002:**
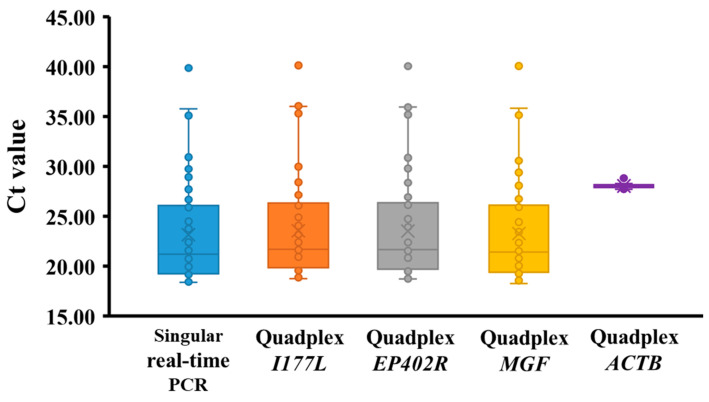
Comparison of Ct values obtained from quadplex and standard singular real-time PCR assays for the detection of ASFV in serum samples from experimentally infected pigs. The box-and-whisker plot, created using Excel 2019, illustrates the distribution of Ct values. The “X” symbol represents the mean.

**Table 1 microorganisms-13-00615-t001:** Primer and Probe Sequences with Corresponding Coverages for Amplifying Target Genes in the *I177L*, *EP402R*, and *MGF360-14L* Gene Databases.

Target Gene (Product Size)	Primer/Probe	Sequence (5′-3′)	Genotypes/Sequence Numbers (n=) and Coverages (%)											Total
			I	II	III	IV	V	VII	VIII	IX	X	XX	XXII	
*I177L* (147 bp)			n = 103	n = 106	n = 1	n = 2	n = 1	n = 3	n = 2	n = 6	n = 5	n = 5	n = 1	n = 235
	Forward1	TGTACTGGAAAAAACTTTATCGG							100%				100%	1%
	Forward2	TGAACTGGAAAAAACTTTAACGG	100%	100%	100%	100%	100%	100%				100%		94%
	Forward3	TGAACTGATATAAATCCTTAACGG								100%	100%			5%
	Reverse1	AATGTGGAAAGATAATGAACAGG							100%				100%	1%
	Reverse2	AATGTGGAAAGTTAATGATCAGG	100%	100%	100%	100%	100%	100%			100%	100%		96%
	Reverse3	AATGTGGAAAATTGATGATAAGG								100%				3%
	Probe	GAAGGGGGATCCGTATAAAATCCTAGCTTG	100%	100%	100%	100%	100%	100%	100%	100%	100%	100%	100%	100%
*EP402R* (145 bp)			n = 108	n = 129	n = 1	n = 2	n = 1	n = 3	n = 2	n = 6	n = 5	n = 5	n = 1	n = 263
	Forward1	ACATGTTGAAGAAATAGAAAGTC	100%	100%	100%	50%	100%	100%	100%	100%	100%	60%		98%
	Forward2	CATGTTGCAGAAATACAAAGTCC				50%						40%	100%	2%
	Reverse1	AGGTGTATTATATTGATAACGACT									60%	60%		2%
	Reverse2	AGGTGTATTATACTGATAACGACT	100%	100%	100%	100%	100%	100%	100%	100%	40%	40%	100%	98%
	Probe	TCTCCCAGAGAACCATTACTTCCTAAGCC	100%	100%	100%	100%	100%	100%	100%	100%	100%	100%	100%	100%
*MGF* *360-14L* (101 bp)			n = 102	n = 105	n = 1	n = 2	n = 1	n = 3	n = 5	n = 6	n = 5	n = 5	n = 1	n = 236
	Forward1	AGAAGACGGGGTTCGGATACAG	100%	100%	100%	100%	100%	100%				100%	100%	93%
	Forward2	AGAAGACGAGATTCGGAGACAG							100%	100%	100%			7%
	Reverse1	GCAAATCCTGAATATGGGCTTATACG	100%	100%				100%	80%			40%	100%	92%
	Reverse2	GCAAATCCTGAATATGGACTTATACG			100%	100%	100%		20%	100%	100%	60%		8%
	Probe1	CCTCCCAGTTCCGCACACAGCCG	100%	100%	100%	100%	100%	100%				100%	100%	100%
	Probe2	CCTCCTAGTTCCGTGCACAGCCG							100%	100%	100%			

Note: “n”, sequence number of each genotype/total number of all genotypes; “%”, percentage coverage of each primer/probe sequence within the corresponding genotype.

**Table 2 microorganisms-13-00615-t002:** PCR Programs for Different Multiplex Reactions.

Path-ID Multiplex One-Step Real-Time PCR Kit	iQTM Multiplex Powermix Kit	QIAGEN Multiplex PCR Kit	Platinum™ Multiplex PCR Master Mix	Multiplex PCR 5X Master Mix
48 °C 10 min 95 °C 5 min 45 cycles: 95 °C 15 s 60 °C 45 s	95 °C 3 min 45 cycles: 95 °C 15 s 60 °C 45 s	95 °C 10 min 45 cycles: 95 °C 15 s 60 °C 45 s	95 °C 10 min 45 cycles: 95 °C 15 s 60 °C 45 s	95 °C 10 min 45 cycles: 95 °C 15 s 60 °C 45 s

**Table 3 microorganisms-13-00615-t003:** Strategy for the Novel Quadplex RT-PCR Assay to Detect ASFV Variants.

Viruses	Probes with Different Dyes			
	VIC-Labeled *EP402R* Probe	FAM-Labeled *I177L* Probe	Texas Red-Labeled *MGF360-14L* Probe	Cy5-Labeled *ACTB* Probe
Wild-type ASFVs	+	+	+	+
ASFVΔEP402R	−	+	+	+
ASFVΔI177L	+	−	+	+
ASFVΔMGF360-14L	+	+	−	+

Note: *EP402R* probe was labeled with 5′-VIC and 3′-BHQ1; *I177L* probe was labeled with 5′-FAM and 3′-BHQ1; *MGF360-14L* probe was labeled with 5′-Texas Red and 3′-BHQ2; *ACTB* probe was labeled with 5′-Cy5 and 3′-BHQ2; “+”, positive; “−”, negative.

**Table 4 microorganisms-13-00615-t004:** Analytical Sensitivity of Quadplex Real-Time PCR with Different Multiplex Reaction Buffers Using Plasmid-spiked Samples.

Target		Quadplex Real-Time PCR with Different Reaction Buffers				
		B1	B2	B3	B4	B5
I177L	R^2^	0.99	0.96	0.96	0.94	0.97
	E	105%	109%	91%	100%	95%
	LOD	1	100	100	10	100
EP402R	R^2^	0.99	0.96	0.94	0.98	0.98
	E	105%	102%	89%	90%	81%
	LOD	10	10	100	100	100
MGF360-14L	R^2^	0.98	0.98	0.98	0.99	0.96
	E	104%	108%	80%	101%	92%
	LOD	1	10	10	10	10

Note: “R^2^”, Correlation coefficient; “E”, PCR amplification efficiency; “LOD”, Limit of detection (Copies/reaction from 200 µL samples); “B1”, Path-ID Multiplex Real-Time PCR reaction buffer; “B2–B5”, four other multiplex reaction buffers tested in this study. The presented data are the average value calculated from three independent experiments.

**Table 5 microorganisms-13-00615-t005:** Sensitivity and Specificity Tests of Quadplex Real-time PCR and Singular Real-time PCR with Virus-spiked Pig Serum Samples.

Viruses		Quantity (TCID_50_) Spiked	Quadplex Real-Time PCR		Standard Singular Real-Time PCR	
			Sensitivity	Specificity	Sensitivity	Specificity
ASFV	OURT88/1 (GI)	Ten-fold serial dilution from 10^5^	LOD = 0.1	+	LOD = 0.1	+
	VNUA-ASFV-05L1 (GII)		LOD = 0.1	+	LOD = 0.1	+
	Georgia strain (GII)		LOD = 0.1	+	LOD = 0.1	+
CSFV	Alfort strain	10^5^	UD	−	UD	−
	C-strain		UD	−	UD	−
PRRSV	VR-2332	10^5^	UD	−	UD	−
	NADC-20		UD	−	UD	−
	JXA1-R		UD	−	UD	−
	1-4-4L1C		UD	−	UD	−
PCV	PCV2b	10^5^	UD	−	UD	−
PRV	Bartha-K61	10^5^	UD	−	UD	−

Note: “LOD”, limit of detection; “UD”, underdetermined; “−” negative; “+,” positive. LODs for all three genes in quadplex real-time PCR are 0.1 TCID_50_. The standard singular ASFV real-time PCR was performed as previously described [42]. The data presented were derived from three independent experiments.

**Table 6 microorganisms-13-00615-t006:** Evaluation of Quadplex Real-time PCR Sensitivity and Specificity Compared to Standard Singular Real-time PCR Assay in Serum Samples from Experimentally Infected Pigs.

Pig Serum Samples	Number of Samples	Standard Singular ASFV Real-Time PCR	Quadplex Real-Time PCR				Positive	Negative	Specificity
			*I177L*	*EP402R*	*MGF*	*ACTB*			
ASFV infected	50	+ (Ct 19–40)	+ (Ct 19–40)	+ (Ct 19–40)	+ (Ct 19–40)	+ (Ct 28)	50/50	0/50	100%
PBS injected	50	−	−	−	−	+	0/50	50/50	100%
CSFV-infected	50	−	−	−	−	+	0/50	50/50	100%
PRRSV-infected	50	−	−	−	−	+	0/50	50/50	100%
PRV-infected	10	−	−	−	−	+	0/10	10/10	100%
BVDV-infected	4	−	−	−	−	+	0/4	4/4	100%

Note: “−” negative; “+” positive. The standard singular ASFV real-time PCR was performed as previously described [42]. The data presented were derived from three independent experiments.

**Table 7 microorganisms-13-00615-t007:** Validation of the Quadplex Real-time PCR with Field Samples.

Samples	Number of Samples	Standard Singular ASFV Real-Time PCR	Quadplex Real-Time PCR				Positive	Negative	Specificity
			*I177L*	*EP402R*	*MGF*	*ACTB*			
Naturally ASFV-infected pig sera	54	+	+	+	+	+	54/54	0/54	100%
ASFV-free pig sera	100	−	−	−	−	+	0/100	100/100	100%
Feral pig sera	6	−	−	−	−	+	0/6	6/6	100%

Note: “−,” negative; “+,” positive. The standard singular ASFV RT-PCR was performed as previously described [42]. The data presented were derived from three independent experiments.

## Data Availability

All data pertinent to the study are included in the article.

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
