# Peer review of "Specific Detection of African Swine Fever Virus Variants: Novel Quadplex Real-Time PCR Assay with Internal Control"

_microorganisms, 2025, doi:10.3390/microorganisms13030615_

Round 1
Reviewer 1 Report
Comments and Suggestions for Authors
The authors reported a quadplex qPCR assay for the detection of African swine fever virus (ASFV). The study is aimed at a highly contagious disease of wild and domestic pigs.
General comments:
Since ASFV is a notifiable disease to WOAH, please mention if approved diagnostic tests are currently available? Since all the positive samples for notifiable diseases, such as ASFV, must be submitted to a reference laboratory, for validation and subtyping/genotyping, what novelty does this assay bring in terms of routine use at a reference or diagnostic laboratory? It is not highlighted if this assay can detect all the reported genotypes?
In the methods, please provide more details on the multiple sequence alignment for primer designing. Please mention how many sequences were downloaded. Which genotypes were included for MSA?
Author Response
Thank you for your thoughtful comments and efforts. Your comments and suggestions greatly helped us improve our manuscript. Below are our point-by-point responses to your comments.
1. Since ASFV is a notifiable disease to WOAH, please mention if approved diagnostic tests are currently available?
Response:
Thank you for your comment. There are several approved diagnostic tests are currently available. We have incorporated following detailed information in the revised manuscript (Pages 2-3, lines 82-98): “There are several approved diagnostic tests for ASFV. The WOAH recommends virus isolation, fluorescent antibody (FAT) testing, and both real-time PCR and conventional PCR. Although virus isolation is the gold standard, its time-consuming nature limits its applicability for real-time disease monitoring. FAT exhibits reduced sensitivity in subacute and chronic ASF. PCR-based techniques, known for their sensitivity, specificity, rapidity, and versatility, are crucial for ASFV detection and are currently the most widely used approach. They enable early detection of the ASFV genome in tissues, EDTA-blood, and serum samples. Numerous commercial PCR kits are available, including the INgene qPPA (Gold Standard Diagnostics), ASFV virus type (Indical Bioscience), ID Gene ASF Duplex (IDvet), RealPCR ASFV (IDEXX), SwineFever combi (Gerbion), ViroReal kit ASFV virus (Ingenetix), and Kylt ASF Real-time PCR (Anicon). The WOAH-certified VetMAX™ African Swine Fever Virus Detection kit (Thermo Fisher Scientific), which underwent the WOAH Biological Standards Commission's registration procedure, is recommended for detecting ASF virus in blood, serum, and tissues from both domestic and wild swine. However, while these kits are generally reliable for detecting wild-type ASFV, they are not suitable for detecting gene-deleted ASFV variants”.
2. Since all the positive samples for notifiable diseases, such as ASFV, must be submitted to a reference laboratory, for validation and subtyping/genotyping, what novelty does this assay bring in terms of routine use at a reference or diagnostic laboratory?
Response:
Thank you for your comment. While confirmatory testing and subtyping/genotyping will still be conducted at a reference laboratory, this assay offers a novel approach for rapid and accurate detection of ASFV variants, specifically targeting variants lacking key genes (I177L, EP402R, and MGF360-14L) in gene-deleted live attenuated vaccines and naturally occurring ASFV variants, which can be missed by routine ASFV detection tests. This rapid initial screening can significantly improve turnaround time for suspect samples and allow for faster implementation of control measures, even before full characterization at the reference lab. It may also be particularly useful in resource-limited settings where rapid initial screening is critical. We added these descriptions in Discussion part of the revised manuscript (Page 11, lines 344-348).
3. It is not highlighted if this assay can detect all the reported genotypes?
Response:
Thank you for this comment. We highlighted that this assay covered all currently available sequences of ASFV I177L, EP402R, and MGF360-14L genes, and genomes containing these genes from GenBank, online ASFV sequence databases, and other published sources manuscript (Pages 3, lines 17-140). The genotypes covered by this assay are showed in the table 1 and described in the Results part (Page 6, lines 226-231): “To construct the I177L, EP402R, and MGF360-14L gene databases, we obtained 235, 263, and 236 sequences, respectively. These sequences represented p72 gene-based genotypes I, II, III, IV, V, VII, VIII, IX, X, XX, and XXII (Table 1). While the majority of sequences were from genotypes I and II, a smaller number originated from other genotypes. Genotypes VI, XI-XIX, XXI, XXIII, and XXIV were excluded due to a lack of available sequences for these genotypes in all three gene databases”.
4. In the methods, please provide more details on the multiple sequence alignment for primer designing. Please mention how many sequences were downloaded. Which genotypes were included for MSA?
Response:
Thank you for your comment. We provided more detail information in Methods part (Pages 3-4, lines 140-144): “Sequence databases for the I177L (n=235), EP402R (n=263), and MGF360-14L (n=236) genes were constructed by combing individual I177L, EP402R, and MGF360-14L gene sequences with sequences derived from the ASFV whole genomes. These databases included genotypes I, II, III, IV, V, VII, VIII, IX, X, XX, and XXII.
Reviewer 2 Report
Comments and Suggestions for Authors
Authors designed a composite sets of probes and a quadplex qRT-PCR for the detection of deletion mutants (of 3 genes) of ASFV and claim that this assay has 100% specificity in their experimental setting. Technically it is good.
Major point: I assume that all data presented in Tables are using all three probes. Line 234, authors claim that gene-deleted mutant will show negative results for one to three probes. I would prefer seeing real data on this exercise, on how to differentiate wild-type ASFV from two LAV vaccines (line 71-77) and the double deletions field isolate (line 78, reference 34).
I suggest delete "sensitive and" from the manuscript title. Here your experimental serum sample come from 7 DPI of unknown infected dose (lines 113-115, line 277), so the sensitivity of this part is unknown. Sera from earlier DPI is more convincing.
delete line 46 and first half of line 47.
Section 2.1: justify why "serum samples" is the only choice for detection. Based on the pathogenesis of ASFV, is their any alternative samples suitable for diagnosis?
line 274, 289: replace "clinical" with "serum".
line 336 and earlier: if this "singular" RT-PCR a recognized "standard", or of its is a "reference" or some sort of control?
Table 1. A total % of coverage should be given for reverse2 primer of MAG.
Author Response
Thank you for your thoughtful comments and efforts. Your comments and suggestions greatly helped us improve our manuscript. Below are our point-by-point responses to your comments.
General comments
Authors designed a composite set of probes and a quadplex qRT-PCR for the detection of deletion mutants (of 3 genes) of ASFV and claim that this assay has 100% specificity in their experimental setting. Technically it is good.
Response:
We would like to thank the reviewer for this positive comment.
Specific points
1. I assume that all data presented in Tables are using all three probes. Line 234, authors claim that gene-deleted mutant will show negative results for one to three probes. I would prefer seeing real data on this exercise, on how to differentiate wild-type ASFV from two LAV vaccines (line 71-77) and the double deletions field isolate (line 78, reference 34).
Response:
We appreciate your insightful comment regarding the presentation of data related to the differentiation of wild-type ASFV and vaccine/field isolate variants. You are correct that demonstrating the assay's efficacy with actual data from ASFV-G-ΔMGF, ASFV-G-ΔI177L, and Asian field isolates with MGF and EP402 deletions would significantly strengthen our findings. While we currently do not have these specific variants in our laboratory, we recognize the importance of this data and are actively pursuing collaborations to obtain them. We intend to conduct these experiments and report the results in a future publication.
2. I suggest delete "sensitive and" from the manuscript title. Here your experimental serum sample come from 7 DPI of unknown infected dose (lines 113-115, line 277), so the sensitivity of this part is unknown. Sera from earlier DPI is more convincing.
Response:
Thank you for your suggestion. We have removed "sensitive and" from the manuscript title.
3. Delete line 46 and first half of line 47.
Response:
Thank you for your suggestion. We have implemented the change and removed line 46 and the first half of line 47, specifically the sentence: “While not directly harmful to humans, ASF's devastating impact on swine populations demands urgent global attention”.
4. Section 2.1: justify why "serum samples" is the only choice for detection. Based on the pathogenesis of ASFV, is there any alternative samples suitable for diagnosis?
Response:
Thank you for raising this important point. Serum was selected as samples for testing and validating of the assay due to their established reliability and common usage in both PCR and ELISA-based ASFV diagnostics. While alternative samples such as oral and nasal swabs are recognized for their potential, viral shedding in these matrices is highly variable and time-dependent, leading to inconsistent results. We acknowledge the value of exploring other sample types and are planning future studies to validate our assay using dried blood spots, oral swabs, nasal swabs, and tissue samples. These findings will be reported in a subsequent publication.
5. Line 274, 289: replace "clinical" with "serum".
Response:
Thank you for this suggestion. We have implemented the requested change and replaced “clinical” with “serum” on lines 274 and 289.
6. line 336 and earlier: if this "singular" RT-PCR a recognized "standard", or of its is a "reference" or some sort of control?
Response:
Thank you for your question. The singular ASFV RT-PCR assay, referenced in line 336 and throughout the manuscript, is a previously published method used as a comparative reference. It functions as a benchmark for the relative performance assessment of our quadplex assay.
7. Table 1. A total % of coverage should be given for reverse2 primer of MGF.
Response:
Thank you for the suggestion. The total percentage coverage for the reverse2 primer of MGF has been added to Table 1.
Reviewer 3 Report
Comments and Suggestions for Authors
The reviewed manuscript is dedicated to the design and validation of a multiplex qPCR-based test for detection of African swine fewer virus, an extremely contagious and dangerous swine pathogen causing significant economic losses worldwide. The manuscript itself is well-written, the methodology is detailed, the study’s design is solid and clear. However, a few minor questions need to be addressed before the possible publication.
- Page 4, line 159: “E. coli” is not italicized
- RT-PCR is an ambiguous abbreviation that can be misunderstood as PCR coupled with reverse transcription. Quantitative PCR seems to be a stricter term.
- A table containing PCR programs for each used master mix would fill the gap of missing PCR conditions.
- Table 3 — a supplementary figure demonstrating the corresponding calibration curves would be highly appreciated.
- Table 5 — a Tukey plot or other similar boxplot is necessary to demonstrate the actual Cq values for each assay.
- Authors are requested to discuss possible study limitations, e.g., possible mismatches under primers or hypothetical “ghost” ASFV strains with all three target genes being altered or deleted.
Author Response
Thank you for your thoughtful comments and efforts. Your comments and suggestions greatly helped us improve our manuscript. Below are our point-by-point responses to your comments.
General comments
The reviewed manuscript is dedicated to the design and validation of a multiplex qPCR-based test for detection of African swine fewer virus, an extremely contagious and dangerous swine pathogen causing significant economic losses worldwide. The manuscript itself is well-written, the methodology is detailed, the study’s design is solid and clear.
Response:
We would like to thank the reviewer for this positive comment.
Specific points
1. Page 4, line 159: “E. coli” is not italicized
Response:
Thank you for pointing out the error. “E. coli” on page 4, line 169, has been corrected and is now italicized.
2. RT-PCR is an ambiguous abbreviation that can be misunderstood as PCR coupled with reverse transcription. Quantitative PCR seems to be a stricter term.
Response:
We appreciate your comment regarding the potential ambiguity of “RT-PCR'”. To address this, we have systematically replaced all occurrences of “RT-PCR” with “real-time PCR” in the revised manuscript.
3. A table containing PCR programs for each used master mix would fill the gap of missing PCR conditions.
Response:
Thank you for the insightful suggestion. In response, we have added Table 2, which now includes the PCR programs for each master mix used.
4. Table 3 — a supplementary figure demonstrating the corresponding calibration curves would be highly appreciated.
Response:
Thank you for your suggestion. Table 3 focuses on presenting the analytical sensitivity, specifically the limit of detection (LOD) and PCR amplification efficiency (E values), of the quadplex real-time PCR across various multiplex reaction buffers. We believe the current data adequately serves this purpose.
5. Table 5 — a Tukey plot or other similar boxplot is necessary to demonstrate the actual Cq values for each assay.
Response:
Thank you for the suggestion. We have added Figure 2, a box and whisker plot comparing Ct values of quadplex and standard singular real-time PCR assays for ASFV detection in serum samples from experimentally infected pigs, to the revised manuscript (Page 9, lines 299-304).
6. Authors are requested to discuss possible study limitations, e.g., possible mismatches under primers or hypothetical “ghost” ASFV strains with all three target genes being altered or deleted.
Response:
We appreciate your comment. We discussed possible study limitations “A potential limitation of this assay lies in the possibility of mismatches due to evolving ASFV strains, including hypothetical variants with significant alterations or deletions within the targeted genes. To mitigate this risk, we will continuously monitor published and GenBank-deposited ASFV sequences, ensuring our primers and probes remain up-to-date and reliable. Furthermore, comprehensive field validation studies, encompassing diverse sample types and environmental conditions, are planned to solidify these promising findings. We will report the findings soon” in the revised version (Page 11, lines 370-377).
Round 2
Reviewer 2 Report
Comments and Suggestions for Authors
The revisions have improved the value of this manuscript.
Author Response
Thank you for your review.
Reviewer 3 Report
Comments and Suggestions for Authors
Many thanks to authors for their thoughtful response and careful correction of the manuscript. All questions were properly addressed and no further editing is needed for the publication of the manuscript.
Author Response
Thank you for your review.